# Identification of errors in draft genome assemblies at single-nucleotide resolution for quality assessment and improvement

Kunpeng Li[1,2], Peng Xu [1,2], Jinpeng Wang[1,2], Xin Yi[1,3] & Yuannian Jiao [1,2,3] ✉

Assembly of a high-quality genome is important for downstream comparative and functional genomic studies. However, most tools for genome assembly assessment only give qualitative reports, which do not pinpoint assembly errors at specific regions. Here, we develop a new reference-free tool, Clipping information for Revealing Assembly Quality (CRAQ), which maps raw reads back to assembled sequences to identify regional and structural assembly errors based on effective clipped alignment information. Error counts are transformed into corresponding assembly evaluation indexes to reflect the assembly quality at single-nucleotide resolution. Notably, CRAQ distinguishes assembly errors from heterozygous sites or structural differences between haplotypes. This tool can clearly indicate low-quality regions and potential structural error breakpoints; thus, it can identify misjoined regions that should be split for further scaffold building and improvement of the assembly. We have benchmarked CRAQ on multiple genomes assembled using different strategies, and demonstrated the misjoin correction for improving the constructed pseudomolecules.

Genome sequencing has vastly improved our knowledge of the genetic bases underlying biological innovations and phenomena. Next-generation sequencing (NGS) and the currently more popular approach, long-read single molecule sequencing (SMS)[1,2], are now routinely used for genome assembly projects[3–7]. The quality of a de novo assembly is influenced by various factors, including read quality, sequencing depth, and the assembler program(s) used[8,9]. However, the quality of a genome assembled de novo is often difficult to precisely evaluate due to the lack of known data[10].

Several approaches are currently used to evaluate the quality of de novo genome assemblies from various perspectives. The N50 contig length is widely used to estimate assembly continuity, but this statistic can be misleading if there are several mis-assemblies of relatively long contigs[11–13]. The Benchmarking Universal Single-Copy Orthologs (BUSCO) program[14] is the state-of-the-art method for evaluation of genome completeness at this time. The approach uses the presence or absence of numerous highly-conserved orthologous genes as a proxy to estimate assembly completeness. However, BUSCO assessments can be inaccurate when the genome in question is a polyploid or recent paleopolyploid, because it is difficult to determine whether part of a subgenome is truly missing or if the assembly is simply incomplete. An arguably better approach to make an informed assessment of assembly quality is to consider the number of real errors in each assembly. QUAST[11,15] compares genome assemblers by estimating assembly errors in contig blocks. This approach requires a known reference genome for the sequenced species or a close relative, meaning that some of the mis-assemblies called by QUAST may be genetic variations rather than assembly errors. Consensus quality (QV)[16] maps short NGS reads mapping back to the de novo assembly to detect errors such as single nucleotide polymorphisms (SNPs) or small insertion-deletions (indels). However, like earlier methods[17–19], this approach is heavily reliant on short-read mapping, which is known to lack alignment accuracy in repetitive or low-accuracy consensus regions[10,20]. A reference-free program, long terminal repeat (LTR)

[1]State Key Laboratory of Plant Diversity and Specialty Crops, Institute of Botany, the Chinese Academy of Sciences, Beijing, China. [2]University of Chinese Academy of Sciences, Beijing, China. [3]China National Botanical Garden, Beijing, China. ✉e-mail: jiaoyn@ibcas.ac.cn

Assembly Index (LAI)[21], gauges assembly quality by estimating the percentage of fully-assembled LTR retroelements (LTR-RTs). LTR-RTs represent a challenge for current sequencing techniques and assembly algorithms; a genome with a low LAI score would be considered poorly assembled. However, LAI underperforms in conducting precise error calls and could be greatly influenced by the dynamic amplification and removal of LTR-RTs in certain species. In addition, several k-mer based approaches, such as JASPER[22], ntEdit[23], KAT[24], Merqury[10], and Merfin[25], have been developed to evaluate assembly accuracy based on differences in k-mers between original high-accuracy sequencing reads and the corresponding assembled sequences. Although k-mer based methods provide single base error estimates, they cannot distinguish between base errors and structural errors.

Genome assemblies often contain errors that range from small nucleotide changes to highly complex genomic rearrangements[8,9,26,27]. Chen et al. developed Inspector[9], which classifies assembly errors as small-scale (<50 bp) or structural collapse and expansion (≥50 bp) errors. Small-scale errors, such as local indels, affect genome accuracy but are often located around repetitive regions, and have a relatively moderate impact on downstream scaffold construction[13]. In contrast, large-scale structural errors (such as misjoined contigs derived from an improper connection of two unlinked fragments) may result in formation of erroneous scaffolds and propagation of errors across multiple scaffolds; this can greatly affect downstream evolutionary or comparative genomic studies[13,28–30]. A key step in resolving large-scale structural errors is to find breakpoints in the problematic contigs and split them at the mis-assembled junctions prior to pseudomolecule construction. Although optical mapping[31] and Hi-C[32] can be used for validation and correction of such errors[13,29,33], both methods perform similarly poorly in their ability to detect misjoins, because they rely on rough inspection of alignments. This approach can only identify approximate conflicting positions and fails to provide the precise locations of misjoined regions.

In the present study, we introduce a new reference-free tool called CRAQ for de novo assembly assessment. CRAQ uses clipping information to reveal assembly errors and low-quality regions by mapping the original sequencing reads back to the draft genome assembly. This enables identification of assembly errors, heterozygous sites, and structural differences between haplotypes at single-nucleotide resolution. By integrating NGS and SMS mapping, CRAQ can identify assembly errors at different scales and transform error counts into corresponding assembly quality indicators (AQIs) that reflect assembly quality at the regional and structural levels. In addition, CRAQ offers the option to correct conflicting contigs by breaking them at relevant error breakpoints; optical maps or Hi-C can then be integrated to fix such errors and improve the assembly.

## Results
### Overview of CRAQ development
Ideally, a high-quality genome assembly should exhibit uniform raw read coverage and few gapped regions or SNP clusters when the original reads are mapped back to the assembly. However, it is common for some assembled regions to show obvious signs of low mapping depth and/or successive base-pair mismatches. The mapping characteristics of these erroneously assembled regions look very similar to the results obtained when reads from individuals with genomic variations are compared to a reference genome. Regions with small-scale local errors typically have no mapped reads or low coverage with typical SNP-cluster features. For regions with large structural assembly errors, such as a misjoin of two genomic fragments, the mapped reads often show characteristics of "clipped reads", a phenomenon in which only part of the read is aligned to the reference. Thus, assessing the mapping status of the original reads along a genome assembly allows assessment of the overall assembly quality and can reveal errors.

We here developed CRAQ, an algorithm that utilizes mapping information from the original NGS short reads or SMS long reads along with the assembled sequences to pinpoint assembly errors at the single-nucleotide level. CRAQ can distinguish between assembly errors and heterozygous loci based on the ratio of mapping coverage and the effective number of clipped reads (Fig. 1, Supplementary Fig. 1). CRAQ classifies putative errors as Clip-based Regional Errors (CREs) or Clip-based Structural Errors (CSEs) depending on the coverage of read mapping and whether there are clipped reads. If a region with clipped NGS reads is spanned by SMS long reads with only SNP cluster features, it is designated as a CRE. If the mapped SMS reads around a region with errors exhibits clipping features (i.e., the NGS reads simultaneously show clipping or no coverage), it is designated as a CSE. The presence of a CSE implies the existence of a misjoin in the genome assembly, which could have significant downstream effects on the usability of the assembly.

We also propose a new genome assembly quality index (AQI), defined as follows:

$$AQI = 100e^{-0.1N/L} \qquad (1)$$

where $N$ represents the cumulative normalized CRE or CSE count and $L$ indicates the total length of the assembly in mega-base unit. To avoid excessive impacts of specific regions enriched in errors (e.g., pericentromeric regions) on the overall AQI values, we normalized error counts within a sliding window of 0.0001 * (total assembly size) (Supplementary Fig. 2), and applied the following equation:

$$Nw = \sum_{i=1}^{m} i^{-1} \qquad (2)$$

where $Nw$ represents the normalized error number in a window and $m$ is the actual number of CRE/CSEs in the block. The assembly qualities of small regions and large structural fragments could be calculated separately as R-AQI and S-AQI.

### Performance estimation with simulations
To benchmark CRAQ performance, we tested the recall and precision on a simulated dataset and compared the results to those generated with the reference-based evaluator QUAST-LG[15] and the reference-free assembly evaluators Inspector[9] and Merqury[10]. We simulated a genome from the human reference assembly (GRCh38) by introducing a total of 11,000 heterozygous variants and 8200 assembly errors (Supplementary Data 2, Supplementary Fig. 3). Heterozygous PacBio HiFi-like reads and Illumina-like reads were simulated using PBSIM[34] and Wgsim[35], respectively (see Methods for details).

With the default settings, the reference-based approach (QUAST-LG) showed the highest F1 score (>98%) in detecting CREs and CSEs among these tested assembly evaluators (Table 1). This is mainly due to we had a perfect reference assembly to compare with. CRAQ identified the simulated heterozygous variants with over 95% recall and precision (Supplementary Fig. 4, Supplementary Data 3); these variants could not be identified by the other assembly evaluators. Notably, CRAQ achieved the highest accuracy among these reference-free programs, with an F1 score (harmonic mean of precision and recall) >97% for simulated errors (Table 1). We also checked these 516 false-negative errors (494 CREs and 22 CSEs) that were not detected by CRAQ, and found that 83.0% CREs and 77.3% CSEs were located in repeat regions (Supplementary Fig. 5a, b). Moreover, for these 516 CRAQ missed errors, there are relatively low or even no reads mapped to these regions (Supplementary Fig. 5c, d and Supplementary Data 4). Inspector had an F1 score of ~96% in detecting CREs, but had low recall (28%) for CSEs. Because Merqury could not distinguish between CREs and CSEs, these errors were merged together, and Merqury had an F1 score of 87.7%. It seems that Merqury failed to identify errors in

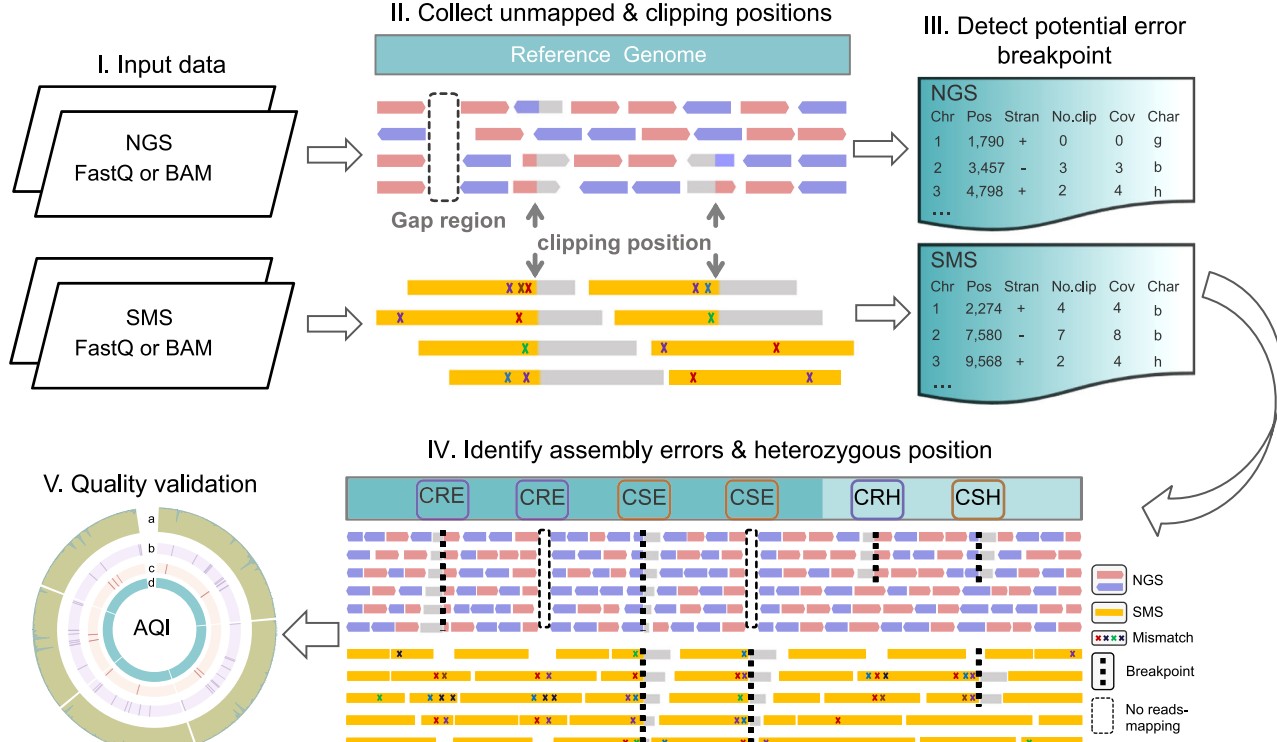

**Fig. 1 | An overview of CRAQ processing steps.** (I) The original next-generation sequencing (NGS) short reads and single molecule sequencing (SMS) long reads are separately mapped to the assembly and the resulting two alignment files are generated after filtering out low-quality reads. (II) Regions with no NGS reads mapped (i.e., gaps) and the positions where NGS/SMS reads are clipped are recorded. The number of clipped reads and the total read coverage at each position are also recorded. (III) For clipped positions from the NGS and SMS alignment, we define heterozygous loci (suffix "h") and the mapping breakpoints (suffix "b") based on a user-defined cutoff for the ratio of the number of clipped reads to the total reads mapped at that position. Together with gaps (suffix "g"), such breakpoints are defined as locations of putative assembly errors. (IV) Putative errors are further classified as Clip-based Regional Errors (CREs) or Clip-based Structural Errors (CSEs) based on read-mapping status. CREs are defined as those with NGS breakpoints or gaps spanned by SMS long reads but enriched in base mismatches; CSEs are defined as those with SMS clipping breakpoints near the NGS breakpoint or gap region. Heterozygous regions are also classified as Clip-based Regional Heterozygosity (CRH) or Clip-based Structural Heterozygosity (CSH) regions based on similar criteria but considering the ratio of mapping coverage. (V) Identified CREs and CSEs are visualized and further used in benchmarking the genome assembly quality. CRAQ outputs a whole-genome summary, regional AQI scores (track a), and the precise location of CREs (track b) and CSEs (track c) for each assembly fragment (track d).

over- or under-assembled repetitive elements due to the lack of additional new k-mer types generated.

## Benchmarking of CRAQ with real datasets

To test the performance of CRAQ on a genome with high heterozygosity, CRAQ was applied to multiple assemblies of an $F_1$ *Drosophila melanogaster* hybrid from a cross of A4 with ISO1[36] (Table 2). The parental genomes were used to distinguish between heterozygous sites and assembly errors. In the HiCanu assembly of the *D. melanogaster* $F_1$ individual, we identified a total of 3006 clipped positions from Illumina reads only and 54 clipped positions from Illumina and PacBio HiFi reads (Supplementary Data 5). After applying CRAQ, we found that only 4.2% (127/3060) of the loci were true assembly errors; 102 were CREs and 25 were CSEs. Moreover, 96% (2904/3006)

of the clipped positions from Illumina reads and 54% (29/54) of the clipped positions from both types of reads were heterozygous loci (Supplementary Fig. 6, Supplementary Data 5). For example, CRAQ identified an assembly error and a heterozygous locus at tig0000001:10,444,000-11,440,000. We compared this contig to the orthologous regions in the parental genomes and examining their read mapping statuses, which confirmed the assembly error in the position x and the heterozygous variant in the position y (Fig. 2).

We further evaluated the performance of CRAQ in identifying large structural errors, comparing its performance to that of the reference-based evaluator Synteny and Rearrangement Identifier (SyRI)[37]. We applied CRAQ and SyRI to publicly-available genome assemblies for *Solanum pennellii* (LYC1722) generated from a single set of Nanopore data with Canu, SMARTdenovo, and Canu combined with SMARTdenovo (CaSM)[38]. The CaSM assembly had the highest assessment score of the available assemblies based on multiple metrics, including BUSCO completeness, LAI, QV score, and N50 contig length (Table 2). We therefore investigated potential assembly errors in the Canu assembly of *S. pennellii* by using CRAQ and SyRI, and the CaSM assembly was used as the reference genome for SyRI. In total, we detected 8029 error related breakpoints using CRAQ, including 7910 CREs and 119 CSEs (Supplementary Fig. 7, Supplementary Data 6), and identified 20,877 SVs (after removing small-scale indels) using SyRI (Supplementary Data 7). To compare these results, we found that ~71.4% (5736/8029) of the errors reported by CRAQ overlapped with 49.8% (6539/13,114) of the SVs identified by SyRI

## Table 1 | Benchmarking error identification of each evaluator with simulation

| | QUAST-LG | | CRAQ | | Inspector | | Merqury |
|---|---|---|---|---|---|---|---|
| | CREs | CSEs | CREs | CSEs | CREs | CSEs | Total |
| Recall % | 98.061 | 98.123 | 95.266 | 96.207 | 95.507 | 28.219 | 84.616 |
| Precision % | 98.957 | 99.112 | 99.763 | 97.942 | 96.750 | 97.283 | 91.091 |
| F1 score[a] % | 98.507 | 98.615 | 97.463 | 97.067 | 96.125 | 43.748 | 87.734 |

[a]F1 score was calculated as F1 score = (2*recall*precision)/(recall + precision). The F1 score was used to measure the accuracy of each evaluator.

**Table 2 | CRAQ metrics and quality statistics for the *Drosophila melanogaster* F$_1$ individual and *Solanum pennellii* genome assemblies**

| Assembler | N50 | BUSCO (%) | LAI | QV | CRAQ | | | |
| --- | --- | --- | --- | --- | --- | --- | --- | --- |
| | | | | | #CRH | #CSH | #CRE (R-AQI) | #CSE (S-AQI) |
| *D. melanogaster* (~150 Mb) | | | | | | | | |
| Peregrine | 12.7 | 99.1 | – | 31.3 | 14.2 | 0.27 | 1.31 (87.7) | 0.047 (95.4) |
| Canu | 13.7 | 99.5 | – | 43.5 | 12.0 | 0.24 | 0.80 (92.3) | 0.053 (94.8) |
| HiCanu | 16.3 | 99.5 | – | 49.3 | 15.1 | 0.21 | 0.71 (93.1) | 0.084 (91.9) |
| Hifiasm | 24.6 | 99.3 | – | 37.8 | 13.8 | 0.25 | 0.66 (93.6) | 0.043 (95.7) |
| *S. pennellii* (~950 Mb) | | | | | | | | |
| Canu- SMARTdenovo | 2.52 | 98.7 | 8.7 | 26.1 | 2.78 | 0.04 | 5.75 (56.3) | 0.119 (88.7) |
| Canu | 1.55 | 98.6 | 7.6 | 24.3 | 3.17 | 0.07 | 9.40 (39.0) | 0.134 (87.4) |
| SMARTdenovo | 1.06 | 98.5 | 7.4 | 22.8 | 3.21 | 0.05 | 10.98 (33.3) | 0.089 (91.5) |

N50 lengths are in mega-bases. "–" represents values that could not be calculated because LAI can only be calculated when the intact and total LTR-RTs contribute at least 0.1% and 5%, respectively, to the genome size. Consensus quality scores (QV) were computed by Merqury. "#CRE/CSE" and "#CRH/CSH" refer to the normalized counts of CRE/CSEs and CRH/CSHs per Mbp.

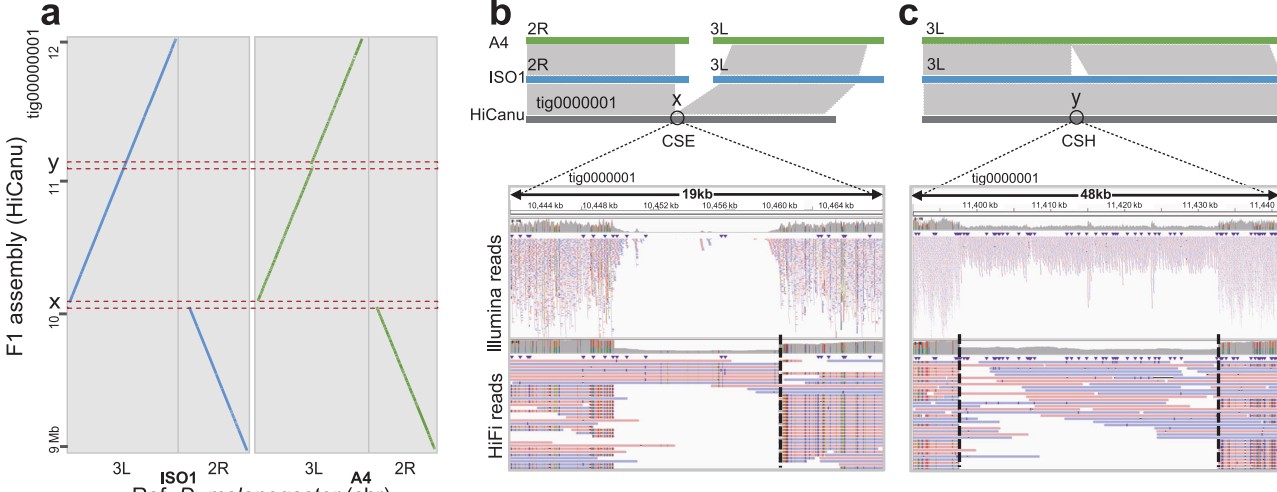

**Fig. 2 | Identification of heterozygous loci and true assembly errors. a** Example of a syntenic comparison by NUCmer between the HiCanu assembly of a *D. melanogaster* F$_1$ individual and the corresponding chromosomes of the two parental strains (paternal, ISO1; maternal, A4). Track x and y on the y-axis represent regions associated with two clipped positions in the HiCanu assembly. Region x and region y were identified by CRAQ as a CSE (**b**) and a CSH (**c**), respectively. The top panel of each graph displays local alignments between the HiCanu contig and references. The bottom panel shows the read mapping status within the CSE (a structural contig misjoin) and the CSH (a heterozygous variant of ~35 kb). The mapping breakpoints are marked with gray dashed lines.

(Fig. 3a, Supplementary Fig. 8). We further investigated the 2292 and 6575 errors uniquely identified with CRAQ and SyRI, respectively. Among the 2292 errors uniquely identified with CRAQ, 56.8% existed in both the Canu and CaSM assemblies (Fig. 3, Supplementary Fig. 9), which could be considered as false negatives for SyRI. Manual inspection suggested that most of the others were also errors in the Canu assembly (see exemplar cases in Fig. 3b, Supplementary Fig. 10). For the 6575 errors uniquely identified with SyRI, there were five main categories: errors in CaSM reference assembly, heterozygous sites, noisy base-error clusters, errors in regions designated as low-confidence by CRAQ, and others (Fig. 3).

To compare metrics produced by the assembly evaluators, we further analyzed 40 publicly-available genome assemblies to characterize the correlations between the R/S-AQI, LAI, QV, BUSCO, and N50 contig length scores (Supplementary Data 1). We found a moderate correlation of R-AQI with other metrics, with LAI having the best correlation ($r^2 = 0.419$) with R-AQI (Supplementary Fig. 11a). Notably, all of the other metrics showed poor correlations with S-AQI (Supplementary Fig. 11b). For example, the SMARTdenovo assembly of *S. pennellii* had the highest S-AQI score (91.5), whereas the CaSM and Canu assemblies

had lower S-AQI scores (88.7 and 87.4, respectively). However, the SMARTdenovo assembly was classified as the assembly with the poorest quality using the other metrics (Table 2). A comparison of the three assemblies to the *S. pennellii* LA716 reference genome[39], demonstrated that the Canu and CaSM assemblies indeed exhibited more structural discrepancies than the SMARTdenovo assembly (Supplementary Fig. 12). Therefore, if the structural quality of the assembly is the primary focus of evaluation, S-AQI values could be superior to other metrics.

## CRAQ identifies misjoined assembly errors for further correction

Contig misjoins often cause severe barriers to scaffolding, and inaccurately assembled scaffolds can lead to misinterpretations in structural genomic studies. CRAQ can separate misjoined contigs at CSE breakpoints, allowing users to reassemble new contigs into scaffolds using Bionano optical maps and/or Hi-C data for correction purpose. For instance, we applied CRAQ to the previously-published *Aquilegia oxysepala* genome[40]. First, draft contigs were generated from the direct output of a de novo assembly of ~50× PacBio sequencing data with Falcon[41] (https://github.com/JiaoLaboratory/CRAQ_data). In total,

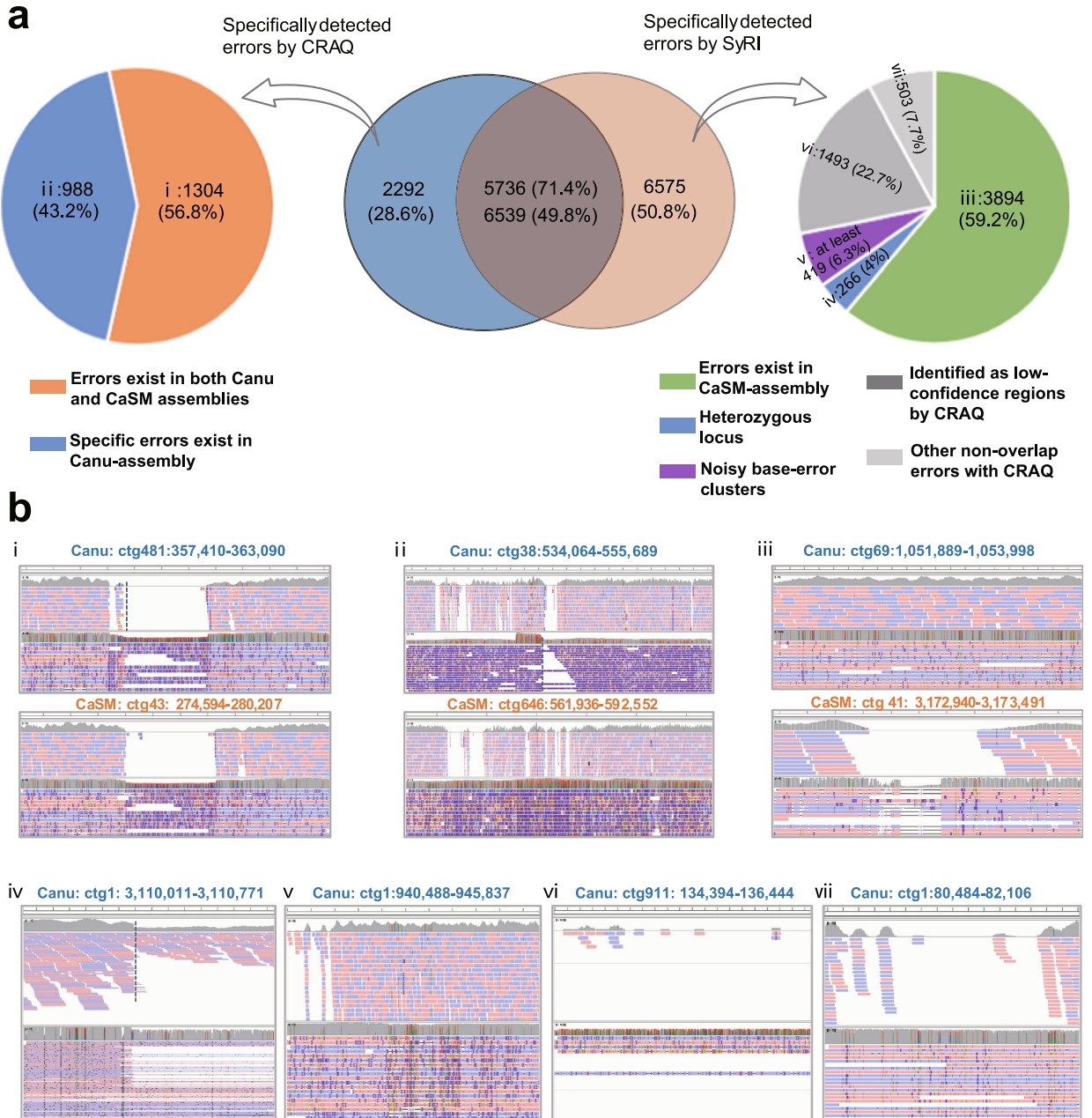

**Fig. 3 | Comparison of errors detected with CRAQ and SyRI in selected *S. pennellii* assemblies. a** Distribution of error types among unique and overlapping errors identified with CRAQ and SyRI. If a SyRI-SV locus fell in a CRE/CSE region, we classified the two errors as the same error. Each category of specific errors identified with CRAQ or SyRI was labeled from 'i' to 'vii', corresponding to data shown in **b. b** Examples of read mapping statuses for error regions of each category labeled in **a**. Pileup plots and coverage from Illumina and Nanopore reads are shown. For tracks 'i', 'ii' and 'iii' the upper panel shows a region in the Canu assembly and the lower panel shows the orthologous region in the CaSM assembly. The mapping breakpoints are marked with gray dashed lines.

we detected 117 CSEs in these draft contigs of *A. oxysepala* (Supplementary Data 8). Using Bionano optical maps and Hi-C data, we generated two scaffold versions: one directly from the draft contigs ("original-scaffold", N50 = 20 Mb, R-AQI = 79.1, S-AQI = 39.0), and the other from CRAQ-assisted split contigs ("corrected-scaffold", N50 = 28 Mb, R-AQI = 78.7, S-AQI = 58.1). We then compared the two versions.

An example case is shown in Fig. 4a, in which contig8 contained a CSE (located at contig8:1,874,290, position y) and was assembled as part of scaffold_3 in the original scaffold version. We referred to contig8 from the beginning position x to y as ctg8_1 and from position y to z as ctg8_2 (Fig. 4a). To confirm whether the contig was an assembly misjoin, we aligned all of the draft *A. oxysepala* contigs to the assembled Bionano optical maps, and found that ctg8_1 mapped to CMAP-1 and ctg8_2

mapped to CMAP-10 (Fig. 4b). Similarly, we observed a bi-partite structure of contig8 (corresponding to ctg8_1 and ctg8_2) in the Hi-C map. Furthermore, ctg8_1 exhibited no contact with the proximal regions of scaffold_3, but a striking contact with scaffold_12 (Fig. 4c). This evidence clearly suggests a mis-assembly of contig8 in the original version. In the corrected scaffold, ctg8_1 and ctg8_2 were assembled in scaffold_11 and scaffold_4, respectively. Ctg8_1 linked downstream of contig30 and ctg8_2 linked upstream of contig70 (Fig. 4d). These contigs were consistent with the optical maps and exhibited no alignment overlap with adjacent contigs (Fig. 4d). There were no anomalous Hi-C contact patterns at the linkage regions (Fig. 4e).

We further compared the genome-wide optical mapping results between the draft and CRAQ-assisted contigs and the Hi-C contact

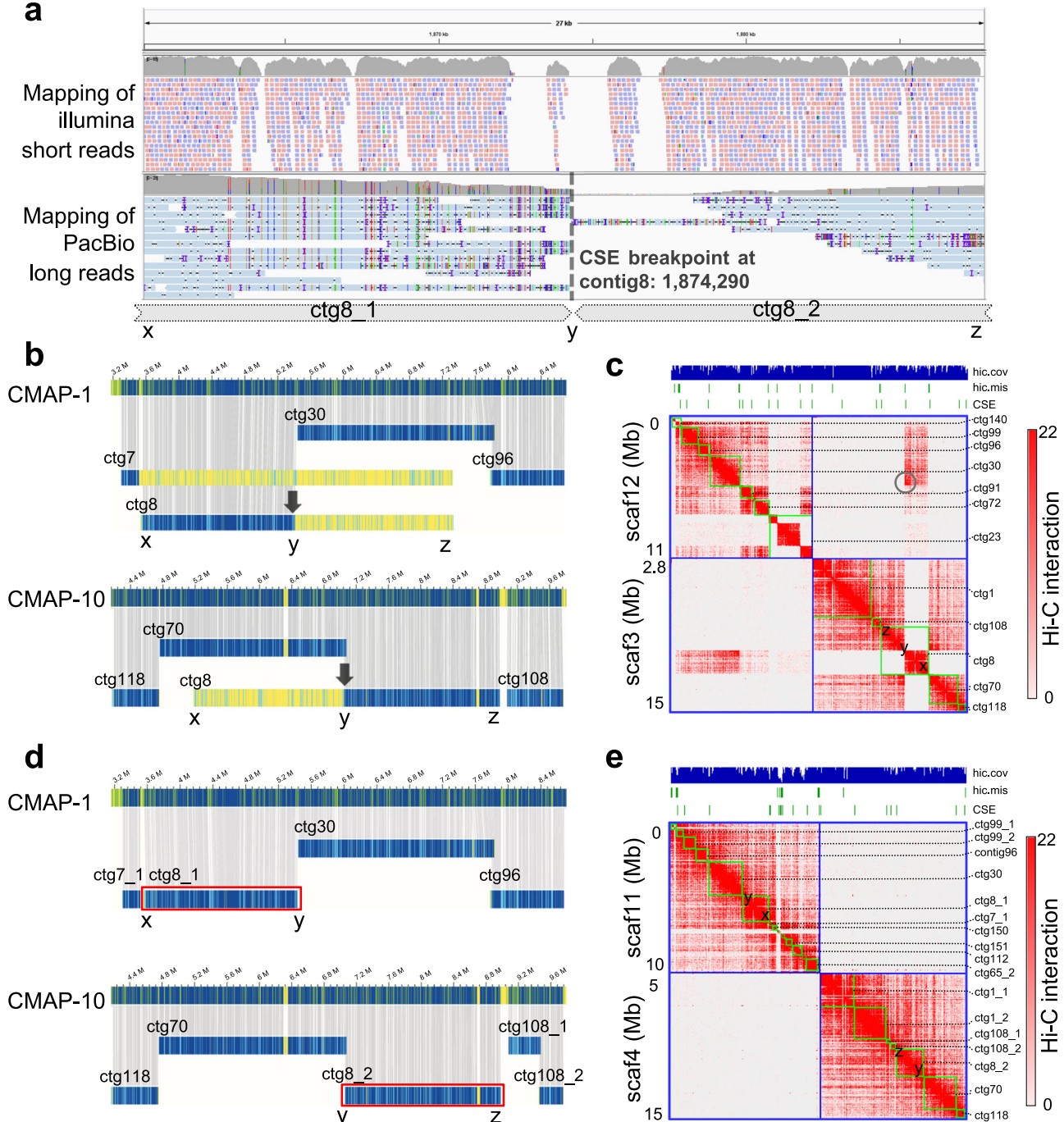

**Fig. 4 | Illustration of CRAQ correction performance on the draft contigs of *A. oxysepala*. a** Mapping status of the Illumina and PacBio reads in a large-scale structural error (CSE) region located on contig (ctg) 8 (3.6 Mbp). The CSE break-point (as determined with CRAQ) is marked with a dashed line, which splits the original ctg8 into two parts, ctg8_1 and ctg8_2. **b** Optical mapping-based alignments of ctg8 with two optical consensus maps (CMAP-1 and CMAP-10). The position of the error is indicated with a black arrow. **c** Local Hi-C contacts of the original scaffolds (scaf). An abrupt depletion of the Hi-C contact signal is observed at position y in ctg8. The gray circle indicates anomalous Hi-C contacts between scaf3 and scaf12. hic.cov (read coverage of Hi-C pairs) and hic.mis (mis-assembly regions) detected with 3D-DNA were also shown at the top tracks. **d** Improved alignments between CRAQ-corrected contigs and optical maps. Ctg8_1 and ctg8_2 are outlined in red. **e** Local Hi-C contacts of the corrected scaffolds. The CRAQ-corrected contigs were re-anchored based on the Hi-C contact data.

patterns between the original and corrected scaffolds of *A. oxyse-pala*. There were 77 misjoin conflicts detected in the draft contigs (Fig. 5a, Supplementary Data 9), indicating severe disagreements between the draft contigs and the optical maps. Moreover, most of the original scaffolds of the *A. oxysepala* assembly exhibited anomalous intra- and inter-scaffold Hi-C patterns (Fig. 5b, Supplementary Data 10). After CRAQ correction, we observed a significantly decreased number of conflicts between the CRAQ-assisted contigs

and the optical maps (Fig. 5c), and a remarkably reduced number of noisy Hi-C signals compared to the original scaffolds (Fig. 5d, Supplementary Data 10). These results indicated that certain genomic regions remained difficult to sequence with high quality, and thus tended to be incorrectly assembled based only on the sequencing reads. It is important to identify, separate, and reassemble these regions based on long-range linking data, such as optical maps or Hi-C contact data.

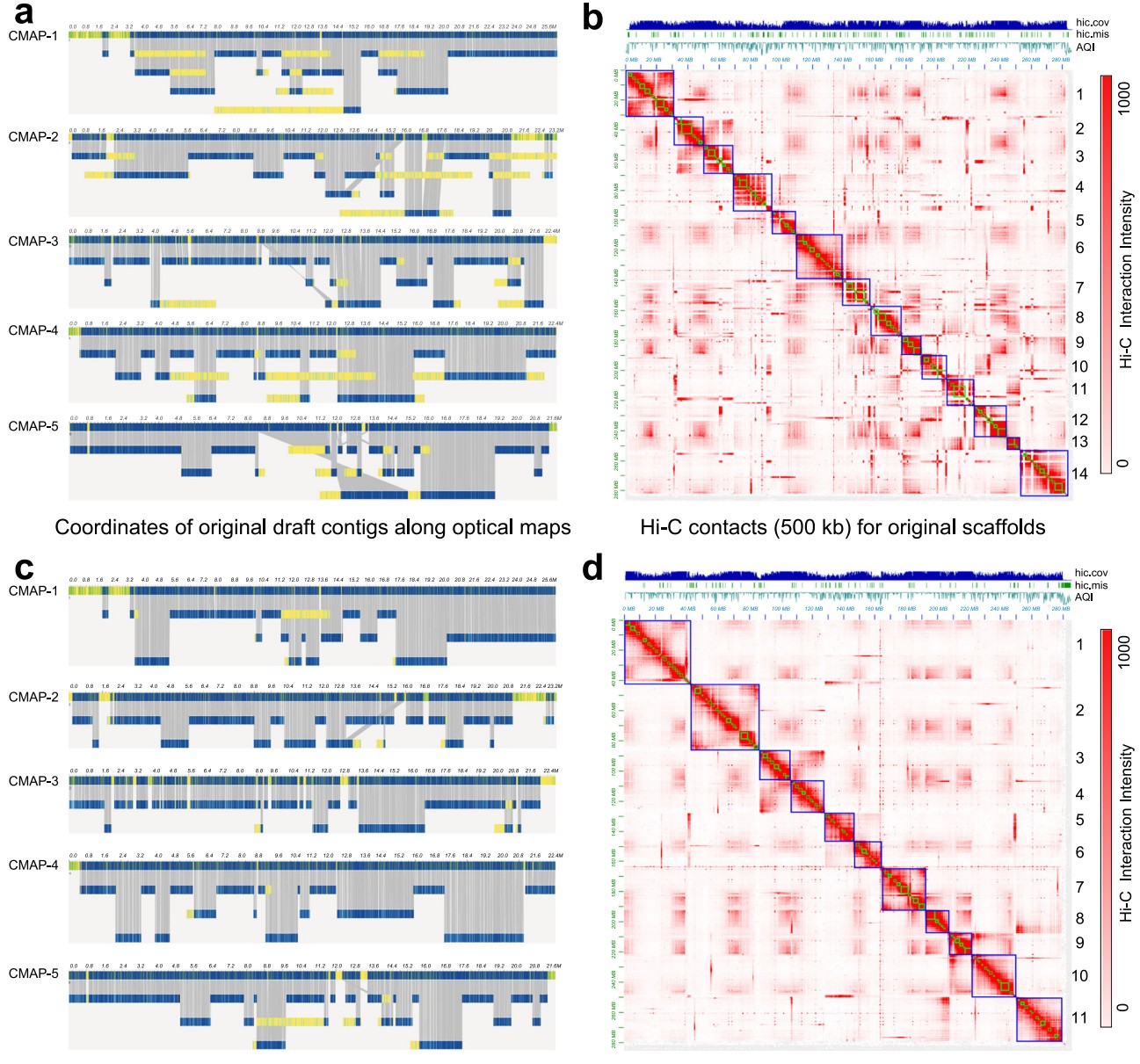

**Fig. 5 | Overview of optical mapping and Hi-C contacts before and after CRAQ correction. a** Alignment of in silico genomic maps, including draft contigs of *A. oxysepala* and the five largest Bionano DLE-1 maps. Collinear DLE-1 markers on the two maps are linked with gray lines. Regions in yellow exhibit breaks in collinearity between the two maps. **b** Hi-C contacts of the original scaffolds of *A. oxysepala*. The order of contigs and scaffolds are presented with green and blue squares, respectively, along the diagonal. The top tracks show the AQI (local AQI score), hic.cov, and hic.mis detected with 3D-DNA. **c** Alignment of in silico genomic maps, including CRAQ-corrected contigs and optical DLE-1 maps. **d** Hi-C contacts of the corrected scaffolds for *A. oxysepala*.

## Discussion

A highly contiguous, accurate, and complete genome assembly is essential for genomic studies, including investigations into chromosome structural variations and evolution of key nucleotides, syntenic analyses, and cis-element predictions. Several well-known tools have been developed to assess genome assembly quality and are widely used to evaluate various parameters of genome assemblies. Traditionally, length metrics (N50/L50 values) provide a standard measure of assembly contiguity. BUSCO[14] and CEGMA[42] are state-of-the-art methods for evaluation of completeness at the gene level. LTR_retriever[43], Merqury[10], and Inspector[9] can be used to evaluate consensus assembly accuracy using LAI and QV values. However, previous evaluators lack consideration of heterozygous loci and provide only a single metric for assembly quality without distinguishing between regional errors and structural misjoins. In the present study, we developed CRAQ, a reference-free genome assembly evaluator, to assess assembly accuracy while considering the heterozygous features of diploid genomes and provide detailed information about assembly errors. These data include the precise locations of CREs/CSEs and both regional and overall AQI metrics for assembly validation.

The inherently heterozygous features of a genome may have strong effects on accurate evaluation of the corresponding assembly when using reads mapping information. However, several previously-developed tools have not implemented heterozygous site removal. We here distinguished between assembly errors and heterozygous regions based on read-mapping coverage data and effective clipping ratio thresholds. Thresholds for these parameters could be defined based on multiple scenarios. We applied CRAQ to highly heterozygous

diploid genomes, demonstrating the accuracy of this tool and the importance of removing heterozygous loci during assembly assessment (Table 1, Supplementary Data 1). The only previously-published tool that considers heterozygosity status is Inspector[9]. In a comparison to Inspector, CRAQ showed much higher performance in distinguishing between heterozygous regions and true assembly errors (Table 1, Supplementary Fig. 13). Identification of heterozygous loci, including CRHs and CSHs, could also help users to better understand the status of an organism at specific loci. In the future, the availability of numerous haplotype-resolved genome assemblies could further resolve such complexity.

Small-scale assembly problems, such as base calling errors or indels, can strongly influence assembly quality. Algorithms such as Racon[44], Nanopolish[45], Medaka (https://github.com/nanoporetech/medaka), and Pilon[46] have been developed to correct inconsistencies associated with base errors or indels with multiple rounds of post-assembly polishing using raw signal data or/and more accurate NGS short reads. Rapid advances in long-read sequencing technologies have greatly improved read accuracy. For example, the availability of PacBio HiFi reads, which are derived from multi-pass sequencing of the same circularized fragment, have achieved per-base accuracy of over 99.9%, comparable to the accuracy of short reads and Sanger sequencing[47]. Assemblies generated from HiFi reads often show high consensus accuracy and do not require read correction[6,47–49]. High-accuracy HiFi long reads can largely eliminate the small-scale inconsistencies discussed above; we therefore primarily focused on assembly errors detected from clipped alignments.

We argue that structural errors in genome assemblies should be attended to and corrected by more researchers, because whole genome-level comparisons will be carried out with increasing frequency in the future to understand chromosome structural evolution, including segmental inversions, translocations, and duplications between or among lineages. Several methods were previously developed to identify these types of errors using reference genomes of closely-related species or several versions of a single assembly[11,15,30]. However, various types of false positives and false negatives likely exist in these circumstances, and it is difficult to distinguish between assembly errors and true structural variations in comparing a newly-assembled genome to a reference assembly. For example, when comparing the CSEs identified with CRAQ to SVs identified in another study[38], we found some CSEs that existed in both the Canu and CaSM assemblies; these errors were therefore overlooked by SyRI[37] when the CaSM assembly was used as a reference to identify SVs in the Canu assembly (Supplementary Fig. 9). In addition, when using a closely-related genome as a reference, false-positive structural errors are likely to occur in the evaluation process due to true structural differences. Therefore, using the original sequencing data from the same species will allow more accurate evaluation of the number of misjoins.

Optical maps and Hi-C contact data have previously been used to detect and correct CSEs[13,33,50]. Bionano optical mapping includes an error correction process, inspecting apparent alignment conflicts between the contig sequence and Bionano maps[51]. Hi-C-based methods split genomic regions for which the contact map exhibits anomalous patterns[52,53]. However, these two approaches often lack the resolution required to precisely identify and split misjoined regions. The Hi-C-based correction approach sometimes yields a higher number of debris fragments due to the aggressive splitting process used[52]. In contrast, CRAQ utilizes read clipping information to conduct error calling, which allows for pinpointing and splitting misjoined regions with single-nucleotide resolution. This method shares a similar underlying philosophy with variant-calling tools such as GATK[54], Freebayes[55], and Deepvariant[56], which were designed primarily for detection of mutational variants using reads from population-scale samples. Further scaffolding these split contigs using optical maps or Hi-C data results in much higher-quality genome assemblies. For

instance, after CRAQ correction, the newly-constructed scaffolds of *A. oxysepala* assembled with Hi-C showed fewer CSE features and thus higher S-AQI values than the original scaffolds (Supplementary Fig. 14, Supplementary Data 10).

We found that misjoined regions were often caused by a very small number of SMS reads that inaccurately bridged two unlinked segments together. These SMS reads frequently showed low sequence complexity or repetitive features and could be multi-mapped back to the misjoined regions (Supplementary Fig. 15). Moreover, specific homopolymer repeats were enriched in CRE and CSE regions (Supplementary Fig. 16). Notably, such multi-mapped reads were filtered out when CRAQ was applied to identify CSE breakpoints by default. Therefore, the current version of CRAQ will perform well for species with monoploid or diploid genomes; evaluation of genome assemblies for species with higher ploidy levels may not be as accurate as the benchmarked cases presented here. Although accurate assessment of polyploid genomes remains a challenge, CRAQ could be expanded for use with polyploid species in the future.

Precise identification of assembly errors remains of paramount importance in accurately assessing genome quality. Our newly developed tool, CRAQ, is a reference-free evaluation method that uses alignment characteristics of the original NGS short reads and SMS long reads mapped back to a genome assembly to validate the assembly quality. After screening out heterozygous sites and structural differences between haplotypes, CRAQ provides precise breakpoint information, assembly error types, and summarized quality scores. In addition, CRAQ offers a correction process to split misjoined contigs at CSEs to aid in accurate scaffold construction. These features of CRAQ facilitate a better understanding of the quality of new genome assemblies and complements existing genome assembly assessment softwares. This tool could be applied to various genome assembly projects to improve assembly quality.

## Methods

### Details of the CRAQ algorithm

**Read mapping and filtering.** The complete framework for CRAQ is shown in Supplementary Fig. 1. CRAQ combines alignment information from NGS short reads (typically from a short insert Illumina library) with SMS long reads (typically from a PacBio CLR/HiFi or ONT library) for genome quality assessment. The pipeline is easy to run, using assembly input files in FASTA format and NGS and SMS sequences in FASTQ/A format. Alternatively, the user can map reads to the assembly in advance and provide two Binary Alignment/Map (BAM) format files as input. In Minimap2 (version 2.18)[57], the '-ax sr' and '-ax map-pb/hifi/ont' options were employed for genomic short-read and different types of long-read mapping, respectively, in CRAQ. SAMTools (version 1.9)[35] was used to convert the alignment files to BAM and to sort the aligned reads. Read mapping is currently the most resource-intensive step of CRAQ. Users could split query sequences into multiple fragments and perform multitasking alignments that would decrease the time required, especially for long-read mapping. Any read alignments with low mapping quality (MAPQ < 20) or that were unmapped, secondary, QC-failed, or PCR-duplicated were filtered out using the '-F 1796 -q 20' parameters in 'samtools view'[35]. If a region in assembly with no or limited coverage after the mapping filter, CRAQ will report these regions as low-confidence regions.

**Extraction of clipped alignments.** The concept of using sequences with clipped alignments has previously been explored for prediction of SVs[58,59]. Here, we adopted this idea by calling genome assembly errors as SV types. CRAQ first extracts all clipped reads, coded as "S" or "H" in the Compact Idiosyncratic Gapped Alignment Report (CIGAR) string from the two filtered BAM (NGS and SMS alignment) files, respectively. CRAQ then identifies the precise base coordinates where clipped reads are mapped and calculates the coverage from clipped reads and total

reads at that position. These data are then used for downstream identification of CRE/CSE breakpoints and heterozygous loci.

**Identification of error breakpoints and heterozygous features.** CRAQ distinguishes error breakpoints and heterozygous loci based on read-mapping coverage data and effective clipping ratios. The clipping ratio thresholds are the fundamental criteria and are calculated as the number of clipping reads divided by the local coverage. Theoretically, heterozygous loci can show an alternative allele in ~50% of clipped reads. However, true assembly error regions lead to a clipping ratio near 100%. The ratio for assembly errors can be lower than 100% in practice due to sequencing errors or inaccurate read mapping, but are still higher than heterozygous regions. By default, a locus is classified as heterozygous when the clipping ratio of NGS/SMS mapping is within a cutoff region h (default = 0.4–0.6). A region is classified as a mapping breakpoint when the ratio exceeds a stringent cutoff value f (default = 0.75). If the assembly regions exhibited coverage over the upper level of h and below the f value, CRAQ reported these regions as ambiguous heterozygous or error region. Together with gaps, such breakpoints are defined as the locations of candidate assembly errors. The filter also excludes candidates with extremely low coverage (m, default = 2) and high coverage (M, default = 5 * average coverage) or poor read mapping quality (SMS clipped length <0.1 * total length) to ensure high confidence of the identified error breakpoints and heterozygous loci.

**Classification of assembly errors.** Assembly errors were classified as CREs or CSEs. CREs are defined as errors in which the SMS long-read spans the NGS breakpoint but has uneven or irregular coverage around the breakpoint. The cutoff for coverage differences is set with the '-d' parameter, which compares the discrepancy in coverage of SMS reads to the 200-bp regions upstream and downstream of the NGS breakpoint with a 20-bp sliding window. CSEs are defined as errors in which the 100-bp region flanking the SMS breakpoint contains an NGS mapping breakpoint or no NGS read coverage. This ensures the correctness of CSE breakpoints because some long reads still suffer from relatively high base error calling and thus incorrect mapping. Additionally, regions adjacent to clipped bases are usually noisy in SMS reads, especially for CSEs. CRAQ can identify the error breakpoint within the noisy region (if the NGS data show a mapping gap) with the option '--error_region'.

**CRE and CSE count normalization.** Some genomic areas, such as pericentromeric regions, are often incorrectly assembled and are enriched in CREs/CSEs. The presence of such regions could greatly decrease the overall AQI value of an assembly. To reduce the weight of such error-prone regions on the overall assembly quality, we normalized CRE/CSE counts by applying Eq. (2): $Nw = \sum_{i=1}^{m} i^{-1}$, where $Nw$ represents the normalized number of CREs or CSEs within a sliding window of 0.0001* (total assembly size) and $m$ is the true number of CREs/CSEs in the block. For example, if three CREs/CSEs were found within one block, the $Nw$ value for that block would be $1/1 + 1/2 + 1/3 = 1.83$. The normalized CRE and CSE numbers were then transformed into the R-AQI and S-AQI scores, respectively. The presence of a CSE implies the existence of a misjoin in the assembly, which could have significant downstream effects on the usability of the assembly. We therefore penalize CSE $Nw$ at a rate 10 times higher than that of CRE $Nw$.

**Quality metric reporting.** CRAQ exports the following output files: (i) a report file that contains the coverage rate of the assembly, the number of CREs/CSEs and CRHs/CSHs, regional AQI scores for each fragment, and summary R-AQI and S-AQI values for the whole genome; (ii) a file with the exact breakpoints of CREs/CSEs and CRHs/CSHs, with supported clipped reads and read coverage information for that error breakpoint or heterozygous locus to facilitate visual inspection in a

genome browser such as IGV[60] or JBrowse[61]; and (iii) a folder that contains identified misjoined fragments (and a newly corrected FASTA file if the user selects the 'correct' function).

**Analysis of simulated heterozygous variants and assembly errors**
To benchmark the evaluation accuracy of CRAQ, we simulated structural and small-scale local assembly errors, as well as heterozygous regions, in the human reference genome hg38 (containing 22 autosomes and an X chromosome)(https://www.ncbi.nlm.nih.gov/datasets/genome/GCF_000001405.26/). Assembled contigs were generated by splitting the genome at "N" bases, excluding fragments shorter than 500 kb. We randomly selected 18,000 genomic loci on the hg38 contigs to simulate heterozygous variants and assembly errors. A total of 11,000 sites were first selected to introduce simulated heterozygous variants, including 10,000 small-local indels and 1000 structural variants. These embedded variants could be considered heterozygous variants. We referred this simulated genome as hg38_sim1. HiFi-like and Illumina-like reads were produced from the original hg38 and hg38_sim1 genomes using PBSIM[34] and Wgsim[35] with the options '--depth 40 --method qshmm --length-mean 10000 --length-sd 2000 --accuracy-min 0.95' and '-e 0.0001 -r 0.0001 -R 0.0001 -s 1 -1 150 -2 150', respectively. These simulated reads would serve as input reads for CRAQ and other assembly evaluators.

To simulate a genome containing assembly errors, we introduced 6000 regional indels and 1000 structural errors (200 fragment indels, 400 contig misjoins, and 400 inversions) at the other previously selected 7000 genomic loci in hg38. Repeat units usually represent the significant impediment to assembly of a new genome, which often cause problems for assembly. Therefore, besides the above 7000 loci, we further introduced 1200 repeat errors, including 1100 small repeat collapses/extensions and 100 large fragment repeats. These repeat loci were randomly selected from the repeat database of hg38 (hgdownload.soe.ucsc.edu/hubs/RepeatBrowser2020/hg38/) and occupied ~10% of the satellite array in hg38. Finally, we generated a simulated error-containing hg38 assembly (referred to as hg38_sim2).

By mapping the above simulated reads to hg38_sim2 genome, we detected errors using CRAQ and other assembly evaluators. These reported errors were further compared with our simulated error type and loci to evaluate the performance of these assembly evaluators

**Genome benchmarking with other metrics and evaluators**
Sources of the sequencing and assembly data used in this study are summarized in Supplementary Data 1. The N50 contig length, BUSCO, QV, and LAI values were calculated separately for each genome. BUSCO completeness was assessed by comparing each genome to a corresponding gene database using BUSCO (version 5.4.6)[14] with the parameters '-lineage path odb10 -mode geno'. For LAI, all LTR-RT candidates were first obtained using LTRharvest[62] with the parameters '-mintsd 4 -maxtsd 6 -motif TGCA -motifmis 1 -similar 85 -vic 10 -seed 20' and LTR_FINDER[63] with the parameters '-D 15000 -d 1000 -L 7000 -l 100 -p 20 -C -M 0.85'. LAI scores were computed based on the identified LTR-RTs using LTR_retriever[43] with default parameters.

Merqury (version 1.3)[10] was used to calculate QV scores and detect errors. Meryl databases were first generated with relevant Illumina reads using a k-mer size of 21 bp. Merqury was then used with each meryl database to evaluate all assemblies with default settings. Merqury identifies erroneous k-mers that are only present in the assembly but not in the input reads. A series of overlapping k-mers were merged into a single error region for benchmarking. An Merqury error was considered validated if the boundary or 21-bp flanking region (one k-mer length) overlapped with the simulated error locus.

Inspector is designed to detect assembly errors with long sequencing reads. This program was used with raw or simulated long reads and the relevant assembly sequences as input and the default

parameters. Inspector identified errors including single SNPs, small indels, regional collapse/expansion, switch errors, and fragment inversions. Small variants (<40 bp) were ignored.

## Detecting structural variations with SyRI

SVs were identified by comparing the reference genomes generated with different assemblers using SyRI[37]. The Canu assemblies (>500k) of *S. pennellii* were input as the query genome and the CaSM assembly was input as the reference due to its higher quality. SyRI output includes SNPs, highly divergent regions (HDRs), deletions (DELs), insertions (INSs), and large fragment misjoins (MJs), all of which were considered to be putative errors in assemblies. Low-quality and SNPs, HDRs, INSs, DELs <40 bp were ignored. Overlapped SV regions were merged. MJ events were further validated through manual inspection. An CRE/CSE was considered to overlap with a SV if the breakpoint fell in the boundary of the SV or within the adjacent 50-bp region.

## Optical mapping for the *A. oxysepala* assembly

Bionano Genomics Direct Label and Stain (DLS) optical consensus maps of *A. oxysepala* were used to identify potential chimeric errors in the draft assembly of *A. oxysepala*. We first performed in silico digestion of the initial *A. oxysepala* draft contigs using the restriction enzyme DLE-1 to produce genomic maps. Subsequently, we applied "RefAligner" (using default parameters) in the Bionano Solve pipeline (version 3.3) (https://bionanogenomics.com/support/software-downloads/) to conduct mis-assembly detection by aligning the optical consensus maps to the in silico maps of the initial *A. oxysepala* contigs. All cuts that conflicted with the optical mapping data were visualized in Bionano Access (version 1.3.0) (https://bionanogenomics.com/support/software-downloads/). The optical mapping-based approach could only infer the approximate genomic locations of misjoins. A CSE breakage was classified as a chimera misjoin if it fell within 20 kbp adjacent to a conflicting optical site; the distance between two nicking enzyme labels was ~10 kbp in our optical molecules. New contigs obtained after breaking these misjoins were re-aligned to the Bionano maps using "RefAligner" as described above.

## De novo scaffolding for the *A. oxysepala* assembly based on Hi-C data

The original and CRAQ-corrected *A. oxysepala* contigs were used as input for the Hi-C scaffolding process. We first employed Juicer (version 1.7.6)[64] to transform the raw Hi-C data into a list of Hi-C contacts with the following parameters: '-s MboI -d juicer -p chrom.sizes -y cut-sites.txt', where file 'cut-sites.txt' was generated using the generate_site_positions.py script. We then performed de novo scaffolding using 3D-DNA (version 180114)[52] based on the generated Hi-C contacts. This program was run without error correction in 3D-DNA using the following parameters: '-m haploid -r 0'. The generated mega-scaffold was only split into scaffolds at large-scale discrepancies in the Hi-C signal near the diagonal. The order and orientation of the generated scaffolds and all anchored input contigs were visualized with Juicebox Assembly Tools (JBAT version 1.8.8)[65].

## Reporting summary

Further information on research design is available in the Nature Portfolio Reporting Summary linked to this article.

## Data availability

The investigated genome assembly data were downloaded from public database, and these corresponding links are provided in Supplementary Data 1. The example data for running CRAQ has been deposited to the repository of GitHub at https://github.com/JiaoLaboratory/CRAQ/tree/main/Example. The human reference genome hg38 were download at https://www.ncbi.nlm.nih.gov/datasets/genome/GCF_000001405.26/. The simulated hg38 genomes used in our project have been deposited in the Zenodo database under accession code https://doi.org/10.5281/zenodo.8383281.

## Code availability

The CRAQ program is available on GitHub at https://github.com/JiaoLaboratory/CRAQ, and at https://doi.org/10.5281/zenodo.8352570, which is free for academic research use.

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

## Acknowledgements

We are grateful to the other members in our laboratory for their suggestions and discussion. We also would like to thank Dr. Zechen Chong (University of Alabama at Birmingham) for sharing the Human (HG002) assemblies from different assembly strategies. This work was supported by the National Key R&D Program of China (2021YFA0909600, Y.J.), the National Natural Science Foundation of China (32221001, Y.J.), CAS

Youth Interdisciplinary Team (JCTD-2022-06, Y.J.), and CAS project for Young Scientists in Basic Research (YSBR-093, Y.J.).

## Author contributions

Y.J. conceived and initiated the project. K.L. conducted the analyses and produced the CRAQ pipeline. P.X. J.W. and X.Y. involved in pipeline testing and script improvement. Y.J. and K.L. wrote the manuscript. All authors read and approved the final manuscript.

## Competing interests

The authors declare no competing interests.
