## [Peer Review File · Nature Communications]

REVIEWER COMMENTS

Reviewer #1 (Remarks to the Author):

In this manuscript, the authors present CRAQ. An assembly error identification tool that is designed to find structural errors in the assembly and report their locations, such that the regions can undergo correction. There is a clear need for such methods for the identification of this kind of assembly errors and the author address a timely problem.

The idea of the method is based on the simple identification of split read alignments (reads that only partially align) in the alignments of the raw read data against the assembly. The usability of the tool however remains unclear as there is no general performance assessment. The manuscript describes anecdotal analysis but lacks general assessment.

In addition, an assembly quality index covering the structural mis-assemblies is presented, which could be a valuable contribution, though there are concerns about its applicability.

Generally, while assembly error correction tools still have huge merit, CRAQ still needs to be shown to be correct in the error identification. Finally, while the manuscript is easy to read and follow, there are several statements that are clearly overselling what CRAQ does and how well it performs. This should be avoided to improve the joy of reading even more.

[Line numbers might not fit exactly as different versions of word tend to rearrange lines.]

Major:

1. The introduction provides an incomplete overview of the current state-of-the-art methods to perform identification and correction of errors in genome assemblies. The authors should include a citation to Merfin, a k-mer based evaluation tool that builds upon Merquy. Additionally, they should discuss how dedicated polishing tools such as Pilon, Arrow, Racon and Medaka, and repurposed variant-calling tools like GATK, Freebayes, and Deepvariant are applied in the context of error correction, and how CRAQ relates to such approaches.

2. The perhaps biggest issue of this work is the lack of assembly performance estimation. It is not clear how many false positive and false negative predictions are made. The manuscript only provides anecdotal evidence; the overall sensitivity and precision of CRAQ with regards how to identify these two types of categories remains unclear. This could be addressed with (realistic) simulations and could address how many patterns are missed (in repeats for example) and how many SER/LER are actually wrongly predicted. The authors do compare CRAQ with Inspector and Syri, but miss to analyze the errors not identified by CRAQ.

For example, using structural variants between the CaSM and Canu assemblies identified by SyRI to evaluate the performance of CRAQ might not resolve the entire performance estimation. A structural variant can be an error in any of the two assemblies, or both can be incorrect, so how can one assess which of the two represents the correct sequence?

Also, even after correction, there seem to be assembly error remaining (yellow bars in Fig5c for example, L-AQI < 60).

Another concern in this direction is the read mapping quality filtering MAPQ<20, which leads to a bias against mildly repetitive regions that cannot be analyzed.

Finally, the comparison to Inspector appears too simplistic. "After comparing to Inspector, CRAQ exhibits much greater sensitivity and specificity during the identification of heterozygous regions and true assembly errors" – Pure count numbers of predictions do not allow to conclude on sensitivity or specificity.

3. Quantification of heterozygosity is too simplistic. The claim that this study demonstrates the accuracy of removing heterozygous loci from assembly correction is not backed up by empirical evidence. The study reports the number of putative heterozygous loci, but no quantitative measure of the sensitivity and precision of CRAQ in detecting such loci. Doubts in this are justified as the definition of het regions seems stringent and would not work for regions with multiple copies or polyploids.

4. AQI: the quality classification implies comparability across species, which is not the case. The

metrics is affected by genome size. With 10k LERs AIQ is ~ 73 in a large genome of size 3.2 GB and 0 in a genome of size 0.1 GB. Hence the classification of the AIQ values into an interpretation of how good an assembly is seems not justified.

5. Exact location of assembly errors: Precise locations are based on alignment clipping. However, regions adjacent to clipped bases are also noisy (Figure 2B, 3A, 4A) and as such it is not proved how the selected clipped location is the precise location for error. Further, all reads do not have clipping at the exact same base (Figure S3), again disputing the claim for "precise location". This needs validation beyond anecdotal evidence of two breakpoints. Further, the overlap analysis with the Inspector/syri output is not clear. How well did the breakpoints overlap?

Minor but required:

1. L130: It needs to be clear to the readers that "small-scale errors (SER)" are in fact not truly all small-scale errors but only patterns that can be found with split reads (which excludes all real small-scale errors). Perhaps a more descriptive name could be found.

2. There is no download link given in the manuscript. The manuscript needs to state the accessibility of the software incl. license, download link etc....

3. I assume that the authors want to imply that the lack of correlation of L-AQI to other quality measures shows that the type of error that is picked up by CRAQ is not included in any of the other measures. I do generally agree with this point, but suggest that this is worked out in the text. It would be helpful to understand the absence/presence of correlation in the individual cases more.

4. In line 550-552: differences between the *S.pennellii* assemblies were regarded as errors in the one generated by Canu. Couldn't these differences be errors in either of the two assemblies?

5. L276: Draft genome assembly correction using CRAQ. CRAQ does not correct the assembly, it helps to find errors, but it does not correct them.

6. Line 370: The claim that CRAQ considers heterozygous variants in polyploid is not demonstrated in this study, as it lacks examples in which CRAQ is applied to polyploid genomes.

7. Line 370: The claim that CRAQ considers heterozygous variants in polyploids is not demonstrated in this study, as it lacks examples in which CRAQ is applied to polyploid genomes.

8. Line 459: The claim that CRAQ outperforms existing genome assembly assessment software is not backed up by the current study, as it lacks experiments that compare the sensitivity and precision regarding the identification of assembly errors of CRAQ with that of current state-of-the-art methods in a quantitative manner. The authors should include such experiments to back up this claim.

9. Line 523: It is not clear why it is a bad thing that errors in pericentromeric regions overly contribute to a reduction in AQI and the number of errors should therefore be normalized for this phenomenon.

Minor:

L141: N is "cumulative normalized count of SER and LER" – what is the normalization in N?

L175: comparison => comparison

Line 207: "We found a moderate correlation of S-AQI with other the metrics, " => delete "the"

Line 281: "For instance, we applied CRAQ to the the previously..." => delete "the"

Line 432: "Therefore, it is worth noting that such multi-mapped reads must be filtered out when CRAQ is employed..." applied?

There were more typos in the manuscript.

Reviewer #2 (Remarks to the Author):

Summary:

Li and coauthors present a novel tool, CRAQ, to address the challenging problem of reference-free evaluation of genome assemblies. While there are several accurate and informative reference-

based evaluation methods, the existing reference-free approaches are mostly qualitative. CRAQ reports assembly quality at single-base pair resolution by taking on input short (e.g., Illumina) and long (e.g., PacBio or ONT) reads and mapping them back to the assembly. The tool detects local and structural assembly errors and distinguishes them from heterozygous sites. Furthermore, CRAQ can correct the assembly by splitting it at most likely misjoins and thus benefit the downstream analysis.

The manuscript is well-structured and includes informative Figures and Supplementary Material. The authors thoroughly benchmarked their software against existing methods and using various datasets. Additionally, Li and coauthors used orthogonal experimental data (optical maps and Hi-C) to demonstrate and validate CRAQ's capability to correct draft assemblies. The tool is freely available on GitHub. Installing and running CRAQ on the sample data provided in the repository was easy. I believe CRAQ may greatly benefit the genomics community especially if the authors consider my comments/suggestions regarding the software. There are also several issues in the manuscript that should be addressed.

Manuscript

Major comments: TODO

- * Since the reference genome is known for some benchmark datasets, CRAQ performance could be compared to reference-based quality assessment methods. E.g., CRAQ's LER/SERs could be compared to the extensive/local misassemblies reported by QUAST (via misassembly coordinates or visually in the Icarus browser).
- * The current text is a bit lengthy which complicates the reading and obscures the scientific value of the manuscript. I suggest shortening some sections and moving very detailed descriptions to the Supplementary Material. This is applicable to sections "Benchmarking of CRAQ and comparison to other assembly evaluation metrics", "Identification and verification of SERs and LERs", "Draft genome assembly correction using CRAQ", and "Discussion".
- * The requirement to have both short and long reads to run CRAQ limits its potential usability. Can the tool potentially work with only one of these data? E.g., at the expense of some reliability or informativity.

Minor comments:

- * lines 488-490: There is a cutoff for detecting heterozygous loci (default = 0.4-0.6) and for detecting mapping breakpoints (default = 0.75). It is unclear how the locations with values 0.6-0.75 are treated.
- * lines 140-141: the AQI formula is not fully intuitive, e.g., why 0.1 was used as the power constant. Also normalized count of SER/LER is explained only in the Methods (lines 507-520), it makes sense to refer to this section from line 141 as it is done in lines 167-168.
- * line 536: using BUSCO (version 3.0.2) -- this version was released almost six years ago, the current version is BUSCO 5 (v.5.0.0 was released in January 2021, the latest is v5.2.1). It is not a direct competitor of CRAQ but it would be good to use the latest versions of software when possible.

Cosmetic/misprints:

- * "respectively" is overused (e.g., lines 87, 171) and also sometimes used incorrectly (e.g., lines 353-355: three tools are "respected" to two value types).
- * Some articles are incorrectly used or missed, e.g., "an mapping" (line 488), "an structural .." (lines 200-201), "these" instead of "the" (lines 160, 187).
- * lines 175: comparation
- * lines 378: assembly -> assemblies

Software

User-friendliness (should be easy to fix):

- * The main CRAQ script requires both the assembly file to analyze (-g Genome.fa) and the file containing its size (-z Genome.fa.size). Since computing the size of a FASTA file is trivial, it could

be embedded directly into the script, so users might provide only one FASTA file.

- * There are many small discrepancies between README and actual filenames, e.g., `Genome.fasta` vs `Genome.fa`, `runAQI` vs `runAQI_out`, `craq.Report` vs `out_final.Report`, `CRAQ/example` vs `CRAQ/Example` (note that the tool is for Linux which distinguishes `E` and `e`).

- * The tool produces three output directories in the current working directory and there is no option to specify a custom output path. Also, the main output directory (`runAQI_out`) contains multiple temporary files (`tmp_*`) that should be removed after the run.

- * Adding the CRAQ output on the example data to the repository would be good. In this case, potential users can directly (without running CRAQ) see what to expect from the tool and whether it would be useful for them.

- * There is no License file, so it is unclear to what extent the tool can be used and/or embedded into other software.

Feature suggestions (more time-consuming but could substantially improve the tool functionality):

- * It would be great to supply the CRAQ output with some graphical representation of the results. E.g., something like Supplementary Figure S5 (stage "V" in Figure 1).

- * The paper says that the CRAQ output can be visually inspected via IGV or JBrowse (lines 526-528). It would be great to supply the GitHub repo with step-by-step instructions with screenshots on how to do this on the example data.

Point-by-Point Response to Reviewer Comments

Reviewer #1 (Remarks to the Author):

In this manuscript, the authors present CRAQ. An assembly error identification tool that is designed to find structural errors in the assembly and report their locations, such that the regions can undergo correction. There is a clear need for such methods for the identification of this kind of assembly errors and the author address a timely problem.

The idea of the method is based on the simple identification of split read alignments (reads that only partially align) in the alignments of the raw read data against the assembly. The usability of the tool however remains unclear as there is no general performance assessment. The manuscript describes anecdotal analysis but lacks general assessment. In addition, an assembly quality index covering the structural mis-assemblies is presented, which could be a valuable contribution, though there are concerns about its applicability.

Generally, while assembly error correction tools still have huge merit, CRAQ still needs to be shown to be correct in the error identification. Finally, while the manuscript is easy to read and follow, there are several statements that are clearly overselling what CRAQ does and how well it performs. This should be avoided to improve the joy of reading even more.

Response: Thank you very much for the time that you have taken to carefully read and evaluate our manuscript. These comments are very helpful and constructive for improving of CRAQ software and the manuscript. We have added new simulation test and other requested analyses and clarifications according to the comments.

[Line numbers might not fit exactly as different versions of word tend to rearrange lines.]

Major:

1. The introduction provides an incomplete overview of the current state-of-the-art methods to perform identification and correction of errors in genome assemblies. The authors should include a citation to Merfin, a k-mer based evaluation tool that builds upon Merqury. Additionally, they should discuss how dedicated polishing tools such as Pilon, Arrow, Racon and Medaka, and repurposed variant-calling tools like GATK, Freebayes, and Deepvariant are applied in the context of error correction, and how CRAQ relates to such approaches.

Response: Thank the reviewer for the good suggestion. Now, we have expanded our introduction and discussion sections by describing and discussing about these other related tools (e.g., Merfin, ntEdit, JASPER) for a more comprehensive overview and comparison. Please refer to modified main text in lines 61-66 and lines 405-409.

In general, tools such as Pilon, Arrow, Racon and Medaka were developed for base-pair or small-indel correction during the assembly process, and here we assume such correction process already performed. Moreover, given the newly developed HiFi technology offering high base-level accuracy reads, we believe that such errors will be very limited and won't be the major concern. Therefore, the main focus of CRAQ is dedicated to identify these relatively large misjoined assembly errors. In addition, these mentioned variant-calling tools such as GATK, Freebayes, and Deepvariant were mainly designed for detecting mutational variant using reads sequenced from population samples. Here, CRAQ mapped the original sequencing reads back to the assembly, and the reads were from one sequencing individual for most cases, and these detected structural variants should be either heterozygous regions or assembly errors. Therefore, CRAQ shared some underlying concept with these mentioned tools, but was developed mainly for a different purpose.

2. The perhaps biggest issue of this work is the lack of assembly performance estimation. It is not clear how many false positive and false negative predictions are made. The manuscript only provides anecdotal evidence; the overall sensitivity and precision of CRAQ with regards how to identify these two types of categories remains unclear. This could be addressed with (realistic) simulations and could address how many patterns are missed (in repeats for example) and how many SER/LER are actually wrongly predicted.

Response: We appreciate the reviewer for giving a great suggestion about adding performance estimation of CRAQ using simulation data, which certainly could be a better way to test the performance of this new tool. We have performed such a simulation test according to the reviewer's idea.

First, we randomly introduced 11,000 heterozygous sites and 8200 assembly errors of small local or large structural errors into the human reference genome (GRCh38). By running the CRAQ and other related tools, we found over 95% of the simulated errors with high precision could be identified by CRAQ (see results in the newly generated Table 1). In addition, just as the reviewer mentioned, about 83% of CRAQ missed and false-detected errors are in the repeat region. In general, the performance of CRAQ is better than Inspector and Mercury, and slightly lower than the reference-based approach of QUAST. We think if there is a perfect reference genome as the simulation study here,

the reference-based approach certainly should be the best choice. But for most cases, it is not possible with such a ground true in hand.

Now, in this revised manuscript, we added a new section "**Performance estimation with simulations**" in the text lines 158-181, which provided simulation details . Thanks again for this constructive suggestion.

The authors do compare CRAQ with Inspector and Syri, but miss to analyze the errors not identified by CRAQ. For example, using structural variants between the CaSM and Canu assemblies identified by SyRI to evaluate the performance of CRAQ might not resolve the entire performance estimation. A structural variant can be an error in any of the two assemblies, or both can be incorrect, so how can one assess which of the two represents the correct sequence?

Response: In addition to the newly added simulation test, we further performed a careful comparison regarding the performance of CRAQ and SyRI on real assembly cases, which could be a good complementary to the simulation test. In the revised manuscript, we have re-write the whole section of this comparison, and generated a new Figure 3, which can clearly show different numbers and relationships of these identified errors in CaSM using these two tools.

We fully agree with the reviewer about that both CaSM and Canu assembly could have assembly errors. In the Canu assembly of *S. pennellii*, we detected 7,910 SERs and 119 LERs (Supplementary Table 5) using CRAQ (Supplementary Fig. 7), and identified 20,877 SVs (after removing small-scale indels) using SyRI (Supplementary Table 6). We found that around 71.4% (5736/8029) of the CRAQ reported errors were overlapped with 49.8% (6539/13114) SVs identified by SyRI (Figure 3a, Supplementary Fig. 8). We further investigated these 2292 and 6575 specifically reported errors from CRAQ and SyRI, respectively. In the CRAQ specifically detected errors, there are 1304 shared mis-assemblies in the CaSM and Canu assemblies, and 988 errors specifically in Canu assembly after manual checking (Figure 3b, Supplementary Fig. 9-10). In the SyRI specifically detected errors, 59.2% (3894/6575) were actually assembly errors in CaSM version. We have revised the related text in the main text accordingly. Please refer to lines 219-232.

Also, even after correction, there seem to be assembly error remaining (yellow bars in Fig5c for example, L-AQI < 60).

Response: We have to admit the correction performance very much relying on the contig assembly quality, optical mapping and Hi-C data. Although the positions of LERs could be identified for splitting the misjoined contigs, it is still

very hard to resolve all of these error places. But we can still see improvements by comparing Fig5c to Fig5a.

In the case of *A. oxyssepa*, although chimeric contigs were split at the positions of LERs, these newly joined regions still exhibited low mapping quality just like 'SNP clusters'. As seen in Supplementary Fig. 14, the newly joined region of ctg8_2 and ctg70 has good Hic contact support, but still exhibits low reads coverage and thusly low AQI. At least for such kind of cases, the low L-AQI values could give us a warning signal if we specifically interested in this region.

Another concern in this direction is the read mapping quality filtering $MAPQ < 20$, which leads to a bias against mildly repetitive regions that cannot be analyzed.

Response: We understand the reviewer's concern. Previously, we suggested this filtering parameter to avoid large number of false positives and false negatives, because it is very important to use properly mapped reads to find the truly clipping signal. This quality control filter could help to exclude those falsely mapped reads, therefore help to identify these regions with abnormal reads mapping coverage. Otherwise, these misjoined regions might show normal reads coverage (Supplementary Fig. 14-15), and could be missed by CRAQ.

But the reviewer is right about that such requirement could potentially lead to some regions that cannot be analyzed (see new Supplementary Fig. 5d). Now, we have further refined CRAQ to report these regions as low-confidence regions if there is no or limited coverage after using the $MAPQ20$ filter. This information will be reported in the output file (runAQI_out/low_confidence.bed). Therefore, we can distinguish whether assembly regions are truly high-quality or low-confidence. We have added this information in Method section lines 458-459. In addition, CRAQ now allow users to try different filtering criteria depending on their main concern about sensitivity or precision.

Finally, the comparison to Inspector appears too simplistic. "After comparing to Inspector, CRAQ exhibits much greater sensitivity and specificity during the identification of heterozygous regions and true assembly errors" – Pure count numbers of predictions do not allow to conclude on sensitivity or specificity.

Response: Now, we have further compared the performance of CRAQ and Inspector using a simulation test, and found Inspector exhibiting much lower recall rate (28%) for structural errors than that of CRAQ (96%). In addition, Inspector could not report heterozygous regions, while CRAQ has over 95% recall and precision (see new Table 1). For more detailed comparison, please refer to "**Performance estimation with simulations**" at lines 158-181.

3. Quantification of heterozygosity is too simplistic. The claim that this study demonstrates the accuracy of removing heterozygous loci from assembly correction is not backed up by empirical evidence. The study reports the number of putative heterozygous loci, but no quantitative measure of the sensitivity and precision of CRAQ in detecting such loci. Doubts in this are justified as the definition of het regions seems stringent and would not work for regions with multiple copies or polyploids.

Response: Now in the newly added simulation test, we have performed quantification of heterozygosity, and added a new Supplementary Fig. 4 for benchmarking of heterozygous variants detection by CRAQ.

About the criteria of defining the het regions, we agree with the reviewer that the h parameter for different genomes (varied repetitive levels) or polyploids (auto- vs allo-) could be very tricky. Here, we have to admit that current version of CRAQ was designed for monoploid or diploid genomes, and a stringent h value of 0.4-0.6 was suggested by default. But users could also try different parameter settings. Now we have added sentences in discussion regarding the complexity and suggested the users to test different threshold range of the heterozygous option (lines 420-423).

4. AQI: the quality classification implies comparability across species, which is not the case. The metrics is affected by genome size. With 10k LERs AIQ is ~73 in a large genome of size 3.2 GB and 0 in a genome of size 0.1 GB. Hence the classification of the AIQ values into an interpretation of how good an assembly is seems not justified.

Response: Sorry, we are not fully understanding the reviewer's point here and probably didn't describe the AQI formula clearly in the previous text. When calculating the AQI values, we have normalized it by considering the genome size. In the proposed formula $AQI = 100e^{-0.1N/L}$, N represents the cumulative normalized count of SER or LER, and L represents the total length of the assembly in mega-base unit. We also The AQI values will be negatively correlated with the density of these identified errors. For the reviewed assumed two scenarios, we think both assemblies suffering high density of structural errors.

5. Exact location of assembly errors: Precise locations are based on alignment clipping. However, regions adjacent to clipped bases are also noisy (Figure 2B, 3A, 4A) and as such it is not proved how the selected clipped location is the precise location for error. Further, all reads do not have clipping at the exact same base (Figure S3), again disputing the claim for "precise location". This needs validation beyond anecdotal evidence of two

breakpoints. Further, the overlap analysis with the Inspector/syri output is not clear. How well did the breakpoints overlap?

Response: We understand the reviewer's concern here. It is hard to confidently determine the exact location of assembly errors. Here we used the breakpoints detected based on clipping alignment to represent the most likely misjoined location of a chimera contig. We agree with the reviewer that when regions adjacent to clipped bases are noisy, flanking regions of the clipping point could also potentially be the misjoined location.

To address this question, we improved our pipeline further and added an option "--error_region" which will lead CRAQ to check the regions flanked the clipped breakpoints. If there is no NGS reads mapping and the SMS reads exhibiting noisy mapping status, CRAQ will report a region together with the clipped location in the output files (locER_out/out_final.CRE.bed & strER_out/out_final.CSE.bed). In addition, we now used this newly reported error region to overlap with these regions reported by Inspector/SyRI, which has been mentioned in the figure legend of Figure 3. In the revised manuscript, we have added some statements in methods for error region identification (please refer to lines 498-501).

Minor but required:

1. L130: It needs to be clear to the readers that "small-scale errors (SER)" are in fact not truly all small-scale errors but only patterns that can be found with split reads (which excludes all real small-scale errors). Perhaps a more descriptive name could be found.

Response: The reviewer is absolutely right about the confusing SER term used here. We only reported error positions that can be found with split reads, which is different from these commonly considered small-scale errors, such as single base errors.

In order to better describe these assembly errors, we changed the names to the Clip-based Regional Error (CRE) and the Clip-based Structural Error (CSE) in CRAQ (see new description in lines 136-138). Similarly, we also defined "CRH" for Clip-based Regional heterozygosity and "CSH" for Clip-based Structural heterozygosity. The previous S-AQI and L-AQI were also changed to R-AQI and S-AQI correspondingly (see new description in lines 155-156).

2. There is no download link given in the manuscript. The manuscript needs to state the accessibility of the software incl. license, download link etc....

Response: Done. We provided the corresponding link (<https://github.com/JiaoLaboratory/CRAQ>), as well as the license, in line 632-633.

3. I assume that the authors want to imply that the lack of correlation of L-AQI to other quality measures shows that the type of error that is picked up by CRAQ is not included in any of the other measures. I do generally agree with this point, but suggest that this is worked out in the text. It would be helpful to understand the absence/presence of correlation in the individual cases more.

Response: Thanks for the suggestion. Now, we have added the case of *S. pennellii* assembly assessment, which reads: “For example, the SMARTdenovo assembly of *S. pennellii* had the highest S-AQI score (91.5), whereas the CaSM and Canu assemblies had lower S-AQI scores (88.7 and 87.4, respectively). However, the SMARTdenovo assembly was classified as the assembly with the poorest quality using the other metrics (Table 2). A comparison of the three assemblies to the *S. pennellii* LA716 reference genome, demonstrated that the Canu and CaSM assemblies indeed exhibited more structural discrepancies than the SMARTdenovo assembly (Supplementary Fig. 12). Therefore, if the structural quality of the assembly is the primary focus of evaluation, S-AQI values could be superior to other metrics”. Please see this newly added description in line250-259.

4. In line 550-552: differences between the *S.pennellii* assemblies were regarded as errors in the one generated by Canu. Couldn't these differences be errors in either of the two assemblies?

Response: Agree. This question has been addressed and responded in the above Major #2 question.

5. L276: Draft genome assembly correction using CRAQ. CRAQ does not correct the assembly, it helps to find errors, but it does not correct them.

Response: Agree. We changed it to “CRAQ identifies misjoined assembly errors for further correction” (line 261).

6. Line 370: The claim that CRAQ considers heterozygous variants in polyploid is not demonstrated in this study, as it lacks examples in which CRAQ is applied to polyploid genomes.

Response: Sorry for the previously inaccurate description. As mentioned in above response, the current version of CRAQ was designed for monoploid or diploid genomes. We have changed this sentence and clearly indicate this in the discussion lines 420-423.

7. Line 370: The claim that CRAQ considers heterozygous variants in polyploids is not demonstrated in this study, as it lacks examples in which CRAQ is applied to polyploid genomes.

Response: This is the same question as the above one.

8. Line 446: The claim that CRAQ outperforms existing genome assembly assessment software is not backed up by the current study, as it lacks experiments that compare the sensitivity and precision regarding the identification of assembly errors of CRAQ with that of current state-of-the-art methods in a quantitative manner. The authors should include such experiments to back up this claim.

Response: As responded above, we have applied CRAQ and other assembly evaluators on a simulation data, as well as some real cases, to compare their performance (main data see Table 1, Figure 3 and Supplementary Fig. 4, 5, 8-10). In addition, we also turned down this statement a little bit, and the current sentence in the main text lines 435-438, which now reads: *“These features of CRAQ facilitate a better understanding of the quality of new genome assemblies and complements existing genome assembly assessment software. This tool could be applied to various genome assembly projects to improve assembly quality.”*

9. Line 508: It is not clear why it is a bad thing that errors in pericentromeric regions overly contribute to a reduction in AQI and the number of errors should therefore be normalized for this phenomenon.

Response: Sorry for the unclear description of the normalization step in previous manuscript. After carefully checking more than 40 genome assemblies in this study (Supplementary Table 1), we found enriched but separated errors in certain regions (see the example below). This could be especially true for peri-centromeric regions. If we simply used these identified errors in the assembly, the overall CRAQ index would be highly influenced by these clustered errors in a short block. Therefore, we feel it is necessary to normalize the number of errors by taking account of their relative positions.

To better understand this normalization, we added a new Supplementary Fig. 2, which shows two scenarios of three errors in two contigs with different relative locations. For one case: the three errors are scattered distributed on the contig, and the total number of normalized errors will be 3. For the other case: the three errors are located close to each other, and the total number of normalized error will be 1.83 ($1+1/2+1/3$). The final R-AQI score will be 74 and 83 respectively. Therefore, the main purpose of the normalization is to reduce the effect of enriched errors in short regions on the final quality score.

In the main text, we have added this description in lines 149-152, which reads: *“To avoid excessive impacts of specific regions enriched in errors (e.g., pericentromeric regions) on the overall AQI values, we normalized error counts within a sliding window of $0.0001 * (total\ assembly\ size)$ ”.*

Minor:

L141: N is “cumulative normalized count of SER and LER” – what is the normalization in N?

Please see the response to Major #9 and the main text lines 149-152.

L175: comparison => comparison

Done.

Line 207: “We found a moderate correlation of S-AQI with other the metrics, “ => delete “the”

Done.

Line 281: “For instance, we applied CRAQ to the the previously...” => delete “the”

Done.

Line 432: “Therefore, it is worth noting that such multi-mapped reads must be filtered out when CRAQ is employed...” applied?

Done.

There were more typos in the manuscript.

Thanks for your careful reading. Now we have gone through the text carefully and edited the language again to solve such kind of problems.

Reviewer #2 (Remarks to the Author):

Summary:

Li and coauthors present a novel tool, CRAQ, to address the challenging problem of reference-free evaluation of genome assemblies. While there are several accurate and informative reference-based evaluation methods, the existing reference-free approaches are mostly qualitative. CRAQ reports assembly quality at single-base pair resolution by taking on input short (e.g., Illumina) and long (e.g., PacBio or ONT) reads and mapping them back to the assembly. The tool detects local and structural assembly errors and distinguishes them from heterozygous sites. Furthermore, CRAQ can correct the assembly by splitting it at most likely misjoins and thus benefit the downstream analysis.

The manuscript is well-structured and includes informative Figures and Supplementary Material. The authors thoroughly benchmarked their software against existing methods and using various datasets. Additionally, Li and coauthors used orthogonal experimental data (optical maps and Hi-C) to demonstrate and validate CRAQ's capability to correct draft assemblies. The tool is freely available on GitHub. Installing and running CRAQ on the sample data provided in the repository was easy. I believe CRAQ may greatly benefit the genomics community especially if the authors consider my comments/suggestions regarding the software. There are also several issues in the manuscript that should be addressed.

Response: Thank the reviewer for the positive impression about our work. We have carefully revised our manuscript and the CRAQ software according to the reviewer's questions and suggestions. These comments are very helpful and constructive for improving of CRAQ software and the manuscript.

Manuscript

Major comments: TODO

* Since the reference genome is known for some benchmark datasets, CRAQ performance could be compared to reference-based quality assessment methods. E.g., CRAQ's LER/SERs could be compared to the extensive/local misassemblies reported by QUILT (via misassembly coordinates or visually in the Icarus browser).

Response: Thanks to the great suggestion. We have compared the performance of CRAQ with that of QUILT using a simulated dataset (Table 1). For more detailed information, please refer to "**Performance estimation with simulations**" at lines 158-181. In addition, we also carefully compared CRAQ

and SyRI (another reference-based tool) performance on assemblies of *S. pennellii*. Please refer to lines 219-232 for more details.

In general, if we have a perfect reference genome, these reference-based quality assessment methods would have better performance than others, such as the simulation test of QUASt (Table 1). But if we don't have a ground truth in hand as references, which is true for most *de novo* genome sequencing projects, the reference-based approaches could lead to large amount of false positive and false negatives. For example, when using the CaSM-assembly of *S. pennellii* as reference to perform error calling in the Canu-assembly, we found that many of these errors are actually from the reference genome, and that some of the true errors are actually in both assemblies(see the new Figure 3, Supplementary Fig. 9).

* The current text is a bit lengthy which complicates the reading and obscures the scientific value of the manuscript. I suggest shortening some sections and moving very detailed descriptions to the Supplementary Material. This is applicable to sections "Benchmarking of CRAQ and comparison to other assembly evaluation metrics", "Identification and verification of SERs and LERs", "Draft genome assembly correction using CRAQ", and "Discussion".

Response: Thanks for the suggestion. We have merged and rearranged some of these mentioned sections, and moved some detailed information to the Methods section. But we also added some descriptions in these sections per the requests of reviewer#1. Hopefully, now this revised version reads smoothly and logically to you.

* The requirement to have both short and long reads to run CRAQ limits its potential usability. Can the tool potentially work with only one of these data? E.g., at the expense of some reliability or informativity.

Response: This is a great suggestion. We have updated CRAQ to allow users only using one of the short or long reads. Please refer to last section "**Running using NGS or long SMS data only**" at <https://github.com/JiaoLaboratory/CRAQ>.

Minor comments:

* lines 488-490: There is a cutoff for detecting heterozygous loci (default = 0.4-0.6) and for detecting mapping breakpoints (default = 0.75). It is unclear how the locations with values 0.6-0.75 are treated.

Response: We thank the reviewer for pointing out this issue. Previously we provided these default parameter settings based on testing dataset, which are relatively strict cutoffs. By default settings, CRAQ could detect both

heterozygous variants and errors in high confidence level. But, just as the reviewer pointed out, these default settings will cause certain amount of locations out of either categories.

Now, we added another output file named “ambiguous.SE.SH and ambiguous.RE.RH” file in the folders of ‘strER_out/ and locER_out/, respectively, which report such locations (e.g. h-values of 0.6-0.75 when running with default settings). We have mentioned this in the Methods lines 481-482. From the tested datasets, we found only a small set of these ambiguous locations. In addition, we further emphasized the settings of these two values in the help page, which should be adjusted according to the sequencing depth and users’ main focus about the assembly evaluation.

* lines 140-141: the AQI formula is not fully intuitive, e.g., why 0.1 was used as the power constant.

Response: In the formula ($AQI = 100e^{-0.1N/L}$), the N represents the normalized number of errors, and L represents the assembled genome size. N/L represents the density of errors (number of errors per Mb). Through evaluating a large set of genome assemblies with different qualities, we classified genomes as following: $N/L \leq 1$ as reference level; $1 < N/L \leq 2.5$ as high-quality level; $2.5 < N/L \leq 5$ as draft-quality level; $N/L > 5$ as low-quality level. When choosing different power settings, the AQI value of genomes in different quality level changes as shown in the following figure.

It seems that 0.1 could be a reasonable value which give AQI value of 90-100 for reference genomes, 78-90 for high quality genomes, 60-78 for draft

quality genomes, and less than 60 for low-quality genomes. Otherwise, if setting power to 0.01, the AQI will be very similar and relatively high values as shown in below table.

Quality-level	Power					
	0.01	0.05	0.1	0.2	0.3	0.4
Reference (N/L						67~10
≤1)	99~10					0
	0	95~100	90~100	81~100	74~100	
High (1< N/L≤						36~67
2.5)	97~99	88~95	78~90	60~81	47~74	
Draft (2.5<N/L						13~36
≤5)	95~97	77~88	60~78	36~60	22~47	
Low (N/L >5)	< 95	< 77	< 60	< 36	< 22	<13

Also normalized count of SER/LER is explained only in the Methods (lines 507-520), it makes sense to refer to this section from line 141 as it is done in lines 167-168.

Response: Thanks to the reviewer's suggestion. Now we have added the normalization information in the main text lines 149-152, which reads "To avoid excessive impacts of specific regions enriched in errors (e.g., peri-centromeric regions) on the overall AQI values, we normalized error counts within a sliding window of $0.0001 * (\text{total assembly size})$ (Supplementary Fig. 2)".

* line 536: using BUSCO (version 3.0.2) -- this version was released almost six years ago, the current version is BUSCO 5 (v.5.0.0 was released in January 2021, the latest is v5.2.1). It is not a direct competitor of CRAQ but it would be good to use the latest versions of software when possible.

Response: Agreed. We have re-run the data using latest BUSCO 5 (v.5.4.6), and the results remain largely the same.

Cosmetic/misprints:

* "respectively" is overused (e.g., lines 87, 171) and also sometimes used incorrectly (e.g., lines 353-355: three tools are "respected" to two value types).

Thanks. We changed these places.

* Some articles are incorrectly used or missed, e.g., "an mapping" (line 488), "an structural .." (lines 200-201), "these" instead of "the" (lines 160, 187).

Done.

* lines 175: comparison

Done.

* lines 378: assembly -> assemblies

Done.

Software

User-friendliness (should be easy to fix):

* The main CRAQ script requires both the assembly file to analyze (-g Genome.fa) and the file containing its size (-z Genome.fa.size). Since computing the size of a FASTA file is trivial, it could be embedded directly into the script, so users might provide only one FASTA file.

Response: We thank the reviewer for the suggestion. Now, the genome size can be computed automatically in the pipeline.

* There are many small discrepancies between README and actual filenames, e.g., `Genome.fasta` vs `Genome.fa`, `runAQI` vs `runAQI_out`, `craq.Report` vs `out_final.Report`, `CRAQ/example` vs `CRAQ/Example` (note that the tool is for Linux which distinguishes `E` and `e`).

Response: Updated. Thanks!

* The tool produces three output directories in the current working directory and there is no option to specify a custom output path. Also, the main output directory (`runAQI_out`) contains multiple temporary files (`tmp_*`) that should be removed after the run.

Response: Thanks for the suggestion. Now, users could specify the output path using the new option of "-D". Also, these temporary files are automatically removed after the run.

* Adding the CRAQ output on the example data to the repository would be good. In this case, potential users can directly (without running CRAQ) see what to expect from the tool and whether it would be useful for them.

Response: Done. We have added the output files on example data to the repository.

* There is no License file, so it is unclear to what extent the tool can be used and/or embedded into other software.

Response: Done. A license file has been added.

Feature suggestions (more time-consuming but could substantially improve the tool functionality):

* It would be great to supply the CRAQ output with some graphical representation of the results. E.g., something like Supplementary Figure S5 (stage "V" in Figure 1).

Response: Thanks for the suggestion. Now, we added an option "--plot", which could plot CRAQ final metrics using Circos plot.

* The paper says that the CRAQ output can be visually inspected via IGV or JBrowse (lines 526-528). It would be great to supply the GitHub repo with step-by-step instructions with screenshots on how to do this on the example data.

Response: Done. We have provided a GitHub repo to show step-by-step instructions with screenshots. Please check out the GitHub link (<https://github.com/JiaoLaboratory/CRAQ/blob/main/Doc/loadIGVREADME.md>).

REVIEWERS' COMMENTS

Reviewer #1 (Remarks to the Author):

Thank you very much for addressing most of my concerns in a very convincing way. While I appreciate the addition of the simulation study, there are a few more points that require some clarifications:

The figures of the simulations do not add up across the manuscript. Supplementary table S2 lists 36746 entries as assembly errors, which does not match the numbers in the main manuscript. There are more ambiguities across the different numbers of this simulation. Another example (line 172), the 516 false negative errors are those CSEs or CREs – this could be made clearer across the paragraph and/or added to table 1.

Line 219 to 232: It is hard to interpret these numbers. That this new paragraph means anything, numbers need to be put in context. The reader would profit from interpretations that the authors have such that it is clear what to learn from them.

Figure3a. It is very hard to understand the details of these pie charts and the description of those. Does the left pie chart not also include false positives? Would those not also add to those? The right pie chart implies that CRAQ does not have any false negatives either? It would help to spend more explanation on this. (Yellow label cannot be read easily.)

Reviewer #2 (Remarks to the Author):

I greatly appreciate the work of the authors on the revision. I believe the manuscript, underlying simulation experiments and benchmarking, the software itself and its documentation significantly improved. The authors have addressed all my concerns, I have only one minor comment/suggestion (see below). The replies to the comments of the second reviewer also look appropriate.

Minor

The new feature to run CRAQ with only NGS or SMS data is highly beneficial for the users, thanks for adding it. However, the corresponding section on the GitHub page ("Running using NGS or long SMS data only" at <https://github.com/JiaoLaboratory/CRAQ>) has a very limited description mostly based on my own comment in the first review ('If the users only have NGS data or SMS long read dta, CRAQ could just take one of these datasets at the expense of some reliability or informativity.'; also mind the 'dta' misprint). From the user's perspective, a more specific description would be much more helpful, e.g., "The lack of SMS data will make this and this metric less reliable", "The lack of NGS data will make this and this metric not informative".

Point-by-Point Response to Reviewer Comments

Reviewer #1 (Remarks to the Author):

Thank you very much for addressing most of my concerns in a very convincing way. While I appreciate the addition of the simulation study, there are a few more points that require some clarifications:

Response: Thanks for the positive feedback about our revision.

The figures of the simulations do not add up across the manuscript. Supplementary table S2 lists 36746 entries as assembly errors, which does not match the numbers in the main manuscript. There are more ambiguities across the different numbers of this simulation. Another example (line 172), the 516 false negative errors are those CSEs or CREs – this could be made clearer across the paragraph and/or added to table 1.

Response: Sorry for the confusion. The listed 36746 entries in Supplementary Data 2 are actually locations of breakpoints, not the simulated errors. Certain type of errors, such as insertions or inversions, have two breakpoints in the original assembly. Deletion errors just have only one breakpoint each. Now, we have modified the *Supplementary Data 2* by adding the simulated error type and IDs before these breakpoints. We also added a new Supplementary Data 4 to show the 516 false negative errors. These two modifications should be able to clarify such confusion.

We also extensively revised the mentioned paragraph, and added more specific reference to figures or tables about these numbers. We thank the reviewer for pointing out these.

Line 219 to 232: It is hard to interpret these numbers. That this new paragraph means anything, numbers need to be put in context. The reader would profit from interpretations that the authors have such that it is clear what to learn from them.

Response: This new paragraph is related to the Fig.3. We have added more interpretations for these mentioned numbers and specifically referenced to the figures, which should clarify the reviewer's confusion here.

Figure3a. It is very hard to understand the details of these pie charts and the description of those. Does the left pie chart not also include false positives? Would those not also add to those? The right pie chart implies that CRAQ does not have any false negatives either? It would help to spend more explanation on this. (Yellow label cannot be read easily.)

Response: Here we used the pie charts to compare the errors detected by CRAQ and SyRI in the Canu assembly of *S. pennellii*. There are 5736 error breakpoints commonly detected by two approaches. 2292 and 6575 error breakpoints were specifically detected by CRAQ and SyRI, respectively. We further investigated these specifically detected errors to better understand the details. In the left pie chart, we find very few false positive for CRAQ. In the right pie chart, we found most of these SyRI detected errors actually false positives, because they are actually not errors in the Canu assembly (specific categories were shown in the right pie chart). Some of these errors in category vi and vii could potentially be true errors in the Canu assembly, and therefore could be potential false negatives for CRAQ. However, they often lack of strong support to be defined as true errors. Given such uncertainty and lacking of ground truth, we prefer to avoid discussing about false positives and false negatives and classify these loci into specific categories in this section.

We also have changed the yellow label to black. Thanks for pointing out this.

Reviewer #2 (Remarks to the Author):

I greatly appreciate the work of the authors on the revision. I believe the manuscript, underlying simulation experiments and benchmarking, the software itself and its documentation significantly improved. The authors have addressed all my concerns, I have only one minor comment/suggestion (see below). The replies to the comments of the second reviewer also look appropriate.

Response: Thanks for the positive feedback about our revision.

Minor

The new feature to run CRAQ with only NGS or SMS data is highly beneficial for the users, thanks for adding it. However, the corresponding section on the GitHub page ("Running using NGS or long SMS data only" at <https://github.com/JiaoLaboratory/CRAQ>) has a very limited description mostly based on my own comment in the first review ('If the users only have NGS data or SMS long read data, CRAQ could just take one of these datasets at the expense of some reliability or informativity.'; also mind the 'dta' misprint). From the user's perspective, a more specific description would be much more helpful, e.g., "The lack of SMS data will make this and this metric less reliable", "The lack of NGS data will make this and this metric not informative".

Response: Thanks for your carefully check and review about our manuscript and software description. Now, we have revised and added more description to the pointed place on the GitHub page, which reads: "If only NGS data or SMS

long read data were available for the sequenced individual, CRAQ could just take one of these datasets as input. However, the lack of SMS long read data will make these CSE and CSH hardly detected. It will also cause more regions classified as low_confidence due to no or limited coverage from NGS data. The lack of NGS data could potentially cause CRAQ report less CRE and CRH, especially for ONT-based assembly.”